# Anti-Inflammatory and Anticancer Properties of Bioactive Compounds from *Sesamum indicum* L.—A Review

**DOI:** 10.3390/molecules24244426

**Published:** 2019-12-04

**Authors:** Ming-Shun Wu, Levent Bless B. Aquino, Marjette Ylreb U. Barbaza, Chieh-Lun Hsieh, Kathlia A. De Castro-Cruz, Ling-Ling Yang, Po-Wei Tsai

**Affiliations:** 1Division of Gastroenterology, Department of Internal Medicine, Wan Fang Hospital, Taipei Medical University, Taipei 116, Taiwan; vw1017@gmail.com; 2School of Medicine, College of Medicine, Taipei Medical University, Taipei 110, Taiwan; 3Integrative Therapy Center for Gastroenterologic Cancers, Wan Fang Hospital, Taipei Medical University, Taipei 116, Taiwan; 4School of Chemical, Biological, Materials Engineering and Sciences, Mapúa University, Manila 1002, Metro Manila, Philippines; leventbless12@gmail.com (L.B.B.A.); marjette.barbaza@gmail.com (M.Y.U.B.); kadecastro@mapua.edu.ph (K.A.D.C.-C.); 5Department of Athletics Sports, College of Humanities and Social Sciences, Chang Jung Christian University, Tainan 711, Taiwan; lyrecojolly@gmail.com; 6School of Pharmacy, College of Pharmacy, Taipei Medical University, Taipei 110, Taiwan; llyang@tmu.edu.tw; 7College of Acupuncture and Oriental Medicine, Houston, TX 77063, USA; 8Department of Medical Sciences Industry, College of Health Sciences, Chang Jung Christian University, Tainan 711, Taiwan

**Keywords:** anti-inflammatory, anti-cancer, sesame extracts, sesame oil, *Sesamum indicum* L.

## Abstract

The use of foodstuff as natural medicines has already been established through studies demonstrating the pharmacological activities that they exhibit. Knowing the nutritional and pharmacological significance of foods enables the understanding of their role against several diseases. Among the foods that can potentially be considered as medicine, is sesame or *Sesamum indicum* L., which is part of the Pedaliaceae family and is composed of its lignans such as sesamin, sesamol, sesaminol and sesamolin. Its lignans have been widely studied and are known to possess antiaging, anticancer, antidiabetes, anti-inflammatory and antioxidant properties. Modern chronic diseases, which can transform into clinical diseases, are potential targets of these lignans. The prime example of chronic diseases is rheumatic inflammatory diseases, which affect the support structures and the organs of the body and can also develop into malignancies. In line with this, studies emphasizing the anti-inflammatory and anticancer activities of sesame have been discussed in this review.

## 1. Introduction

Numerous studies have already been able to prove that different kinds of food are capable for acting against several types of diseases due to their various medicinal properties. *Sesamum indicum* L., more commonly referred to as sesame, is one of the foods that are known to exhibit pharmacological applications. Sesame, from the Pedaliaceae family [1], has been known way back several decades ago (1600 BC), and is originated from Indonesia according to the Hindu legends. Sesame is considered as one of the oldest condiments in the human history [2,3]. In line with this, sesame is one of the first crops to produce oil [4]. Its herbal medicinal property was first discovered in China and India during the 8th century BC, and is believed to relieve toothaches, to give energy, to prevent aging, to sooth the mind and body and to treat bites of insects [5,6]. The largest area of sesame cultivation is found in India and provides 27.9% of the world’s sesame [7]. Some countries also cultivate sesame especially in the tropical countries of Africa and Asia [8].

Majority of the sesame lignans are found in the sesame seeds. Sesame seeds comprise of 50% oil and 25% protein and the rest are sugars, moisture, fibers and minerals [9,10]. There are four commonly known sesame lignans namely, sesamin, sesamol, sesaminol and sesamolin. These sesame lignans (Figure 1) are known for its various biological activities and applications. Sesamin, a furfuran lignan, plays a role in scavenging free radicals and lipid and glucose metabolism. Sesamol possess high antioxidant activity, which involves membrane protection from peroxidation of lipids [11,12]. Sesaminol exhibits inhibitory property against membrane lipid peroxidation and improves the tocopherols availability of vitamin E through enhancement of concentrations of liver and plasma [13,14]. Sesamolin upregulates the rate of peroxisomal fatty acid oxidation and hepatic mitochondrial and has a synergistic effect on pyrethrum insecticides [15,16,17]. Not only is sesame oil abundant in sesame seeds, but sesame oil also has excellent quality nutrition and stability [18]. Among other vegetable oils, sesame oil has many uses and has high therapeutic values, which make it interesting to study. The presence of sesame lignans such as sesamin, sesamol, sesamolin and other acylglycerols (oleic acid, linoleic acid, palmitic acid, stearic acid and arachidic acid) in sesame oil contributes to its distinct properties [19,20]. This paper presents an in-depth review on the anti-inflammatory and anti-cancer properties of sesame. 

## 2. Pharmacological Applications of Sesame

### 2.1. Anti-Inflammatory Activity

Inflammation has been recorded way back several centuries ago (1500 BCE–600 CE) in the ancient Indian medicine called Ayurveda [21]. In the modern era, the term ‘inflammation’ comes from the Latin word inflammare that has a meaning of “to set on fire” [22]. Redness, heat, pain and swelling are the characterizations of inflammation and are induced by several factors such as frostbite, infection by pathogens, burns, chemical irritants, physical injuries, oxidative stress, ischemia, toxins and hypersensitivity [23,24]. Inflammation is part of the body’s defense mechanism against pathogens or pathogen-associated molecular patterns [25]. Restoring, regenerating and repairing of the damaged tissues or organ hemeostasis through cellular network and signaling pathways are involved in the inflammation process. Conversely, severe acute inflammation or prolongation of inflammation may lead to pathology, organ failure, chronic inflammatory diseases, autoimmunity and death. The active cellular components that are responsible for the processing of acute and chronic inflammation are monocytes/macrophages, endothelial cells, neutrophils, innate lymphoid cells, mucosal-associated invariant T cells, mast cells, natural killer cells, dendritic cells and other lower forms of T cells [26,27,28,29]. Table 1, Table 2 and Table 3 depicts the in vitro/in vivo models for the anti-inflammatory properties of sesame.

#### 2.1.1. Neurodegenerative Disease

Dysregulation of microglia, a principal cell type of brain, is pivotal to recruitment of cell inflammation and expression of pro-inflammatory factors, which later results to neurodegeneration [30]. Such neurodegenerative diseases are Parkinson’s disease, Alzheimer’s disease and other multiple sclerosis [31]. The inflammation begins when the pattern recognition receptors (PRRs) and pathogen associated molecular patters (PAMPs) interacts [32]. A common type of PPRs that detects the lipopolysaccharide (LPS) is Toll-like receptor 4 (TLR4), which is expressed highly on the surface of the microglia when activated. Once activated, the pro-inflammatory mediator is released via mitogen-activated protein kinase (MAPK) and nuclear factor kappa-light-chain-enhancer of activated B cells (NF-кB) pathways [33] and further generation of other pro-inflammatory mediators begins (e.g., interleukin 1 beta (IL-1β), interleukin-6 (IL-6), tumor necrosis factor alpha (TNF-α), reactive oxygen species (ROS) and nitric oxide (NO)). Other pro-inflammatory molecules like prostaglandin E_2_ (PGE_2_) are also initiated simultaneously through phosphorylation of its subunits. There are several NF-кB subunits such as p-IKKα/β, p-IκBα and p-p65, and several MAPK subunits such as c-Jun N-terminal kinase (JNK), p38 and extracellular-signal-regulated kinase (ERK).

In the literature study [34] showed that wherein the BV-2 microglial cell line was used, sesamin shows dose-dependent decrease in expression of TLR4 in LPS-stimuli. Notably, at 50 μM concentration of sesamin, there is a significant decrease in the expression of TLR4. Furthermore, sesamin suppresses the phosphorylation of p-IkBα and p-p65 over a moderate period. In addition, sesamin decreases the phosphorylation of JNK, however, the phosphorylation of p38 is slightly reduced. Interestingly, sesamin with 50 μM indomethacin dose-dependently reduced *IL-1β* and *IL-6* mRNA gene expression and moderately reduced TNF-α before LPS exposure. Similarly, sesamin decreases the levels of cyclooxygenase-2 (*COX-2*) gene expressions and hinders the production of PGE_2_. Moreover, sesamin reduced the expression of inducible nitric oxide synthase (*iNOS*) gene and as well as reduced production of NO in dose-dependent manner. Sesamin exhibits a reduced effect for neurotoxicity of LPS-mediated microglia activation, which eventually increases the viability of neuronal cells.

In another study [35], in which same microglial cell line was used, further investigation on the p38 MAPK signaling pathway was commenced to support its role in cytokine production. It has been found that 50 μM concentration of sesamin suppresses the p38 MAPK activation (40–75%) induced by LPS. The inhibitory effect of sesamin is similar to that of SB203580 (p38 MAPK inhibitor), which inhibits the production of IL-6 mRNA and protein production specifically. Similarly, sesamolin has been reported to reduce the activation of p38 MAPK induced by LPS, however it has not yet been fully studied.

One of the mental disorders associated with neuroinflammation is depression. The downregulation of norepinephrine (NE), and serotonin (5-hydroxytryptamine, 5-HT) levels and the decrease of synaptic content cause depression [36,37]. Brain derived neurotrophic factor (BDNF) controls the development of neuronal function and becomes ineffective in neurodegenerative disorders such as depression [38]. The ionized calcium binding adaptor molecule 1 (IBA-1) is responsible for the microglia activation in hippocampus and cortex [39]. Several studies already used chronic unpredictable mild stress (CUMS) for the physiological pathway elucidation of depression [40].

CUMS-induced depression in mice [41] shows relatively positive results toward forced swimming, tail suspension, elevated plus maze, Morris water maze and Y-maze tests when sesamin is administered (50 ppm/d) for 6 weeks. Furthermore, sesamin upregulates the levels of 5-HT and NE in striatum only, suggesting its beneficial effects on depressive like behaviors. Neurotrophin-3 (NT3) and BDNF shows increase expression in hippocampus when treated with sesamin. Comparatively, sesamin decreases the expression of IBA-1 expression and as a result, the production of inflammatory cytokines ceases.

Another neurodegenerative disorder is the ischemic brain stroke. Shutting off the flow of cerebral blood to thrombi results to ischemic brain stroke due to loss of oxygen and energy supply to crucial tissues of the brain. In middle cerebral artery occlusion (MCAO)-treated mice [42], sesamin (30 ppm) mitigates brain injury by suppressing the production of inflammatory mediators. Interestingly, sesamin reduces the expression levels of p-ERK1/2 together with p-p38 of ischemic brain tissue in MCAO-induced brain damage.

Sesamin at dosage 20 ppm has shown to suppress 6-hydroxydopamine (6-OHDA) that induces Parkinson’s disease in rats [43] via decrease in inflammatory mediator levels in the brain. Sesamin has the capability to alleviate astrogliosis based on the lowering effect of glial fibrillary acidic protein (GFAP) immunoreactivity. Similarly, the inhibitory effect of sesamin against inflammatory agents (MAPK and *COX-2*) results to stabilizing the oxidative stress and mortality in kainic acid-induced status epilepticus [44]. This study, however, is not fully studied as the degree of inflammatory markers was not discussed.

Degradation of heme to iron, carbon monoxide and biliverdin is caused by a phase II antioxidant enzyme called heme oxygenase (HO). There are other isoforms of *HO* such as inducible HO-1 and constitutive HO-2 and HO-3 [45]. Biliverdin reductase transform biliverdin to bilirubin and bilirubin is believed to possess anti-oxidative properties [46,47].

At 100 μM concentration of sesamin, there is an increase in the (*HO-1*) protein level in RAW 264.7 macrophage cells. The *HO-1* mRNA expression, however, is not affected by sesamin. Several studies have claimed that sesamin activates Nf-кB or MAPK signaling pathway, not to mention, sesamin affects the p65 and p38 MAPK effectively in RAW 264.7 cells. The ZnPP IX, a HO-1 inhibitor, reduced the inhibitory effect of sesamin on the release of NO. The degradation of HO-1 protein through the ubiquitination pathway is partially suppressed by sesamin. The proteasome activity, however, is not affected by sesamin. Hence, the ubiquitination mechanism inhibition by sesamin is still unclear [48].

In rat pheochromocytoma PC12 cells study [49], episesamin and sesamin metabolites are investigated. Based from the luciferase reporter assays, episesamin and sesamin metabolites has the capability to activate the signaling cells nuclear factor E2-related factor 2/antioxidant response element (Nrf2/ARE), which further upgrades to phase II detoxification enzyme expression. Moreover, sesamin metabolites induced the expression of detoxification enzymes such as HO-1, y-GCSc and NQO-1 in a dose-dependent manner. Different signal transduction pathways are also affected by the sesamin metabolites through phosphorylation. In addition, sesamin metabolites increase the expression of HO-1 mRNA and protein, which comes before the nuclear translocation of Nrf2.

In the 16-week-old senescence-accelerated mouse-prone 8 (SAMP8) study [50], sesaminol exhibit a reducing effect on the inflammatory cytokines namely IL-6, IL-1β and TNF-α via real-time polymerase chain reactor (PCR) assay in the brain of the modeled mouse. This further contributes in the mitigation of the Alzheimer’s disease.

#### 2.1.2. Osteoarthritis

The degradation and the tearing down of cartilage matrix are characterized in a chronic articular disease called osteoarthritis (OA) [51,52]. Chondrocyte has become the context of pathogenesis of OA, which has an outcome of imbalance between degradation and synthesis of cartilage extracellular matrix (CEM). Inflammatory cytokine network could stimulate matrix metalloproteinases (MMPs) generation together with PGE_2_ and NO in chondrocytes [53,54]. During the progression of OA, MMPs (e.g., collagenase-3/MMP-13) cleaves and denatures the type II collagen and proteoglycan at the surface of the cartilage. The degradation of the collagen and proteoglycan results to the loss of tensile strength in the matrix of cartilage due to the increase of water content [55,56]. Interleukin-1 (IL-1) and TNF-α can prompt the expression of MMPs expression via c-fos signaling pathway.

In papain-induced OA rat study [57], sesamin has been reported to exhibit chondroprotective effects for OA. Sesamin has no effect on the activity of aggrecanase, the main proteoglycan in the cartilage tissue, but has reversible effect in the expression of MMP-1, -3 and -13 in the human articular chondrocyte (HAC) culture. Not to mention, sesamin suppresses the IL-1β induced MMP expression at both protein and mRNA levels. Moreover, sesamin has been reported to reverse the synergistic effect of combined IL-1β and oncostatin M (OSM) and stops the degradation of type II collagen and proteoglycan. Having said this, sesamin can slow down the destruction of cartilage prompting to the development of OA.

Another study [58], wherein the articular cartilage of the 12 patients undergoing knee replacement surgery were acquired, suggests that sesamin activates Nrf2 signaling pathway and up-regulates HO-1 protein expression and further inhibits the inflammatory gene expressions in the OA chondrocytes. Coupled with, sesamin inhibits the activation of NF-кB during the process of Nrf2 activation.

Sesamol exhibits an inhibitory effect on MMPs expression that triggers OA [59]. The specific MMPs are MMP-1, -9 and -13 and the expression of these MMPs initiate the destruction of cartilage. The cell line used was the human chondrosarcoma cell line (SW1353) and was put into different concentrations of sesamol (5–20 μM). The study claimed that activation of MMP-9 and its expression is prompted concentration-dependently by TNF-α. MMP-1 and MMP-13, however, is not improved by TNF-α but still these are inhibited by sesamol in response to PMA (phorbol 12-myristate 13-acetate). Sesamol, moreover, has been reported to inhibit the degradation of IкB-α, which also considers the inhibition of NF-αB activation, which then propagates the TNF-α signaling. In restoring the chondrocytes via IL-1β signaling pathway, there would be production of MMP-9. The IL-Iβ induced p38 MAPK activation promotes generation of MMP-9 by chondrocytes. Having said this, sesamol has been found to alleviate the phosphorylation of p38 MAPK induced by IL-1β. Moreover, sesamol has been shown to attenuate the expression of MMP-1/-9 in MIA-induced OA in rats though the effect of sesamol in the destruction of cartilage was not studied.

One study [60] discussed the effectivity of encapsulating sesamol with micelles. Encapsulated sesamol in phosphatidyl choline micelles have no cytotoxic activity in the cells and improves its bioavailability against inflammatory. As compared to free sesamol, encapsulated sesamol has a higher percentage in decreasing the production of ROS induced by LPS intracellular. Free sesamol decreases the generation of ROS by 42.6%, whereas encapsulated sesamol decreases the generation of the same species by 74.8%. Investigation on the extent of encapsulated sesamol on other inflammatory cytokines is not yet discussed.

In a clinical trial study [61], a randomized double arm, double-blind active-controlled was designed in 104 male and female participants ages 30–70 years old. Half of the participants are the control group and half are the intervention group. The intervention and took 1.5 mL of sesame oil thrice a day for a span of 4 weeks, while the control group took diclofenac gel in the same span of time. The clinical results show that sesame oil reduces knee OA pain and other body systems related to knee OA more effectively than the control group. On the other hand, the control group was non-inferior as compared to sesame oil when it comes to knee joint stiffness. The study claimed to be the first clinical trial to conduct efficacy of sesame oil in OA patients. All things considered, the study needs extensive and deep exploration regarding other external and internal factors of OA and its response to sesame.

#### 2.1.3. Liver Disease

Nonalcoholic fatty liver disease (NAFLD) occurs when there is generation of fat in the liver in the absence of alcohol. NAFLD is characterized by lobular hepatitis, hepatic steatosis, and liver cell injury [62]. One of the mechanisms involved in the pathogenesis of NAFLD is the inflammation in hepatocytes [63]. Nuclear receptor such as Liver X receptor α (LXR-α) and peroxisome proliferator-activated receptor α (PPAR-α) are responsible for the regulation of lipid homeostasis and are crucial [64]. 

In the high-fat diet (HFD) Sprague-Dawley rat model [65], sesamin administered at dose 160 ppm by weight exhibit a reduced level of steatosis through inhibition of gene tags: *TC* and *TAG* accumulation. Equally, sesamin reversed the effect of HFD to the levels of inflammatory cytokines such as IL-6 and TNF-α. In parallel, sesamin downregulates the LXR-α prompted by HFD, while sesamin upregulates the expression of PPAR-α, which stimulates attenuation in the hepatic fat accumulation.

The life threatening of fulminant hepatic failure (FHF) is described as the immense hepatocyte necrosis and sudden decrease in the liver function. This results to multiple organ disfunction syndromes and hepatic encephalopathy [66,67]. In lipopolysaccharide/d-galactosamine (LPS/d-GalN)-induced FHF mice model [68], 100 ppm of sesamin shows improved mortality and promotes serum aminotransferases activity for 6 h. Moreover, sesamin possess hepatoprotection by reducing the regulation of expression of hepatic *TNF-α* mRNA and protein as well as intercellular adhesion molecule-1 (ICAM-1) and endothelial cell adhesion molecule-1 (ECAM-1). Similarly, sesamin downregulates the expression of TLR4 hindering the MAPK and NF-кB signaling pathway. In one small study, sesamin alleviates the synergistic effect of combined lead (Pb) and LPS causing acute hepatic injury effectively. The suppressive effect of sesamin positively affects several inflammatory signaling pathways such as *COX-2*, *iNOS*, JNK and p38 MAPKs, CHOP and GADD45b [69].

The conversion of sesamin into catechol derivatives and converted further into glucuronides or sulfates in the liver [70,71,72,73] has been reported to have anti-inflammatory effect on macrophage cell line (J774.1 cells). The said sesamin metabolites have been reported to have strong inhibitory effects on the LPS-induced NO production [74]. Through methylation, the catechol group of the sesamin metabolites was found to cause this inhibition. Though the level of the inhibitory effect of the sesamin metabolites differs, its inhibitory strength is relative to one another. One of the noticeable interactions is the deconjugation of (7α,7′α,8α,8′α)-3,4-dihydroxy-3′,4′-methylenedioxy-7,9′:7′,9-diepoxylignane (SC1) glucuronide to (7α,7′α,8α,8′α)-3-methoxy-4-hydroxy-3′,4′-methylenedioxy-7,9′:7′,9-diepoxylignane (SC1m) in macrophages. With this, the time-dependent deconjugation enhances the inhibitory effect against the LPS-induced NO production in macrophages. Correspondingly, the inhibitory effects of SC1 glucuronide are controlled by the β-glucuronidase activity and catechol-*O*-methyltransferase (COMT) activity in macrophages. β-glucuronidase activity hinders the accumulation of SC1m in macrophages and hence, withdraws its inhibitory effect. On the other hand, the COMT activity helps the inhibitory effects of SC1 glucuronide in macrophages. The sulfates of the sesamin metabolites, however, possess weak or non-inhibitory effects towards the LPS-stimulated macrophages. The conversion of sesamin to SC1 during the metabolism of macrophage is therefore the prominent form for the anti-inflammatory effects [75].

Sesame oil, at dosage 1 and 2 ppm, possess protective effect against nutritional steatohepatitis in the methionine-choline deficient (MCD) mice model [76]. Sesame oil with concentration of 4 ppm has been reported to mitigate hepatic injury, α-SMA, fibrosis and reduces the activity of MMP-2, and -9. However, at the same concentration of sesame oil, it elevates the expression of PPAR-γ and tissue inhibitor matrix metalloproteinase 1 (TIMP-1) [77]. It was mentioned, consequently, that consumption of a large dosage of sesame oil might cause steatohepatitis due to suppression of its antioxidative effect [78]. PPAR-γ, when activated, is responsible for suppressing the expression of collagen and inhibition of hepatic fibrosis by blocking the profibrogenic transforming growth factor beta (TGF-β)/Smad pathway which then down-regulates the expression of α-SMA expression [79,80,81]. MMPs cleave the fibrillar extracellular matrix (ECM) and aid in apoptosis, specifically MMP-2 and -9. As mentioned above, sesame oil inhibits MMP-2 and -9 as it regulates the expression of TIMP-1. This results to alleviating the injury, hepatic stellate cell (HSC) activation and degradation of ECM and utmost is reversing the fibrosis [82]. Conversely, the US Food and Drug Administration’s approval for its medication for the treatment of non-alcoholic steatohepatitis (NASH) has not yet been made.

In a different rat model [83], sesame oil has been found to possess preventive expression of proteins in HFD-induced endoplasmic reticulum (ER) stress and to prevent the initiation of apoptosis. Furthermore, sesame oil has been reported to decrease the expression levels of lipogenic transcription factors (by 44.58%) and enzymes in the liver, which also increases the expression of fatty acid oxidation-related gene. Similarly, sesame oil suppresses the regulation of sterol regulatory element-binding protein 1 (SREBP-1) by 22.91% and the expression of FAS by 50.38%, however, it promotes the expression levels of PPAR-α and carnitine palmitoyl transferase 1 (CPT-1). In view of these regulations, this increases the hepatic lipid oxidation and hence prevents hepatic fat accumulation.

#### 2.1.4. Diabetic Eye Disease

Diabetic retinopathy is a microvascular complication that affects the neurovascular of the retina as a result of neurodegeneration, neuro-inflammation, eventual fibrosis and other diabetic-related damage [84]. In general theory, pathological conditions related to inflammatory activates microglia in the retina and further initiate neuro-inflammation [85]. This eventually led to tissue ischemia, vascular occlusion and cell death causing blindness [86].

In streptozotocin (STZ)-induced diabetic retinopathy mice model, administered 30 ppm of sesamin alleviates the retinal inflammation. The treatment decreases the mRNA levels of *TNF-α* and *ICAM-1* I unlike in the diabetic group. This further elaborates the possibility of reducing the production of inflammatory cytokines in diabetic retina. Sesamin suppresses the induced diabetic retinal injury by inhibition of TNF-α and microglia Iba-1 [87].

#### 2.1.5. Inflammatory Bowel Disease

Ulcerative colitis is an inflammatory bowel disease (IBD) that involves damaging of mucosal tissue via dysregulation of the inflammatory system. In the dinitrochlorobenzene (DNCB)-induced IBD albino rat model [88], sesamol decreases the activity of myeloperoxidase (MPO), which is considered to manifest the anti-inflammatory activities. Although sesamol has incapability to decrease the IL-6 and TNF-α cytokine levels induced by DNCB, sesamol can undergo through ROS pathway.

Aspirin, when ingested, can kill different inflammatory diseases however this can possibly cause acute gastroduodenal injury, which considers the bleeding of ulcers [89]. This happens when there is lipid peroxidation in gastric mucosal caused by ROS, which then regulates the inflammatory cytokines [90,91]. In an aspirin-induced gastric mucosal rat model [92], sesamol has been reported to have suppression effect on neutrophil activation and infiltration when aspirin induces gastric inflammation. The activation of neutrophil initiates the expression of proinflammatory mediators and eventually upregulates the production of nitric oxide, which causes cell damage and lipid peroxidation. The inhibition of sesamol on the neutrophil activation, moreover, is reported to have no effect on the physiological aspect of the aspirin-treated system.

In one study [93], sesaminol triglucoside can be metabolized to enterolignans through the walls of the large intestines such as ST-1, ST-2 and ST-3. Correspondingly, these sesaminol triglucoside metabolites are transformed further into hydroxylate metabolites when absorbed to the intestines and is excreted in urine. ST-1 and ST-2 possess catechol moiety, which considers its antioxidant activity. ST-2 has showed a remarkable inhibitory effect against the LPS stimulated TNF-α and IL-6 production in RAW 264.7 cells. STG blocks the generation of NO, which is induced by LPS and inhibits cytosolic phospholipase A2 (cPLA2), *COX-2* and *iNOS* expression [94]. 

In one small study of adhesive small bowel obstruction (SBO) [95], a clinical trial wherein sixty-four patients (control: 33 patients; intervention: 31 patients) were administered, is conducted in a span of three hours with 150 mL sesame oil. The results show more effectivity of sesame oil in SBO as compared to the control group. Only a few patients were required to undergo surgery in the intervention group as compared to the control group and the observed duration of stay in the hospital was shorter than the control group.

#### 2.1.6. Cardiovascular Disease

Endothelial dysfunction ignites the start of the chronic inflammatory process called atherosclerosis. This pathogen pathway is enhanced by transforming low-density lipoprotein (LDL) to oxidized low-density lipoprotein (oxLDL) as a consequence of ROS and oxidative stress. Exposure of endothelial cells to oxLDL promotes the expression of proinflammatory cytokines [96].

In the human umbilical vein endothelial cells (HUVECs) model [97], sesamin at a dose of 100 μM has nearly 100% inhibitory effect on the activation of NF-кB induced by oxLDL. Similarly, sesamin also suppresses the discharge of interleukin IL-8, and endothelin ET-1, prompted by oxLDL. The expression of adhesion molecules on the surface has been attenuated by sesamin by half of the initial dose.

Cardiac hypertrophy is the unusual enlargement of heart muscles that leads to the changes in the extracellular matrix of the heart. Renin-angiotensin system (RAS) is crucial in the development of cardiac hypertrophy especially in the left ventricle. MAPK stimulates the hypertrophic response of Angiotensin II (Ang II), the bioactive peptide component of RAS [98,99,100,101]. In the study of DOCA/salt-induced left ventricular hypertrophy (LVH) rat model, sesame oil exhibited a down-regulation of p-p38 and p-JNK levels. However, sesame oil had no observed effect on the ERK activation in the LVH rat [102].

#### 2.1.7. Lung Disease

Acute lung injury is caused by lung inflammation as a consequence of endotoxemia [103]. Promotion of pulmonary inflammatory cell sequestration and enhanced production of pro-inflammatory mediators affects the alveolar space resulting to lung dysfunction [104,105,106,107]. In the study of systematic endotoxin-induced acute pulmonary injury in rats [108], administered sesamol at 1–3 ppm blocks the inflammatory cells induced by LPS in infiltrating the alveolar space, suppresses the protein leakage and expression of inflammatory cytokines in bronchoalveolar lavage fluid (BALF). The NF-кB activation in alveolar macrophage is inhibited by sesamol. Sesamin at dosages 1 and 3 ppm, moreover, suppresses the generation of nitric oxide related to the alveolar macrophage.

Leukotrienes are mediators of lipid that involves the pathogenesis in asthma. The enzyme 5-lipoxygenase (5-LOX) draw in the production of inflammatory mediators through metabolism of arachidonic acid into leukotriene B_4_ (LTB_4_) and leukotriene C_4_ (LTC_4_), potent inflammatory mediators, by the presence of five lipoxygenase activating protein (FLAP). LTC_4_ is claimed to possess powerful inflammatory eicosanoid, which enhances vascular permeability [109]. Sesamin has been reported to reduce the levels of LTB_4_ more effectively than sesamol. On the contrary, sesamin and sesamol downregulates the serum level of LTC_4_. The cumulative effects of the combined sesamin and sesamol reduces the said inflammatory mediators, although its reducible strength is equivalent to that of the individual effects of sesamol and sesamin suggesting that the synergistic interactions are absent. For the most part, sesamin and sesamol exhibits anti-leukotriene effects, which downregulates the receptors and key enzymes of leukotriene pathway and further diminishing the pro-inflammatory leukotrienes production [110].

Non-heme iron-containing enzyme called lipoxygenases (LOX) is characterized by its catalysis activity in incorporating molecular oxygen into polyunsaturated fatty acids [111]. The conversion of arachidonic acid to hydroxyeicosatetraenoic acid (HETE) via metabolism with LOX promotes expression of leukotrienes and regulates the inflammation pathway [112].

In the study of kinetic inhibition [113], sesamol competitively inhibits LOX with IC50 value of 51.84 μM and an inhibitory constant (K_i_) of 4.9 μM. The competitive inhibition is happening in either the active substrate site or the active metal ion site. Hence, in the ferric reducing ability power (FRAP) assay of the same study, 55.35 μM of sesamol reduces half of the Fe^3+^-LOX into Fe^2+^-LOX indicating the partial interaction of sesamol to the active metal ion site.

**Table 1 molecules-24-04426-t001:** In vivo models for the anti-inflammatory effects of sesame lignans.

Compound	Inflammatory Disease/Disorder	Rat Model	References
Sesamin	Depression	Chronic unpredictable mild stress (CUMS) rat model	[41]
	Ischemic brain stroke	Middle cerebral artery occlusion (MCAO) rat model	[42]
	Parkinson’s disease	6-hydroxydopamine (6-OHDA) rat model	[43]
	Osteoarthritis	Papain-induced osteoarthritis rat model	[57]
	Hepatic steatosis	High-fat diet rat model	[65]
	Fulminant hepatic failure	d-galactosamine (d-GalN)-sensitized rat model	[66]
	Acute hepatic injury	Lead-induced acute hepatic injury rat model	[69]
	Diabetic Retinopathy	Streptozotocin (STZ) induced rat model	[87]
	LPS-induced leukotrienes generation	ad libitum semi-synthetic diet rat model	[110]
Sesamol	Ulcerative colitis	Dinitrochlorobenzene (DNCB)—induced rat model	[88]
	Gastric ulceration	Aspirin-induced gastric mucosal rat model	[92]
	Acute lung injury	Endotoxin-induced acute pulmonary inflammation rat model	[108]
Sesaminol	Alzheimer’s disease	senescence-accelerated mouse-prone 8 (SAMP8) model	[50]
Sesame Oil	Nonalcoholic steatohepatitis	Methionine-choline deficient (MCD) diet rat model	[76]
	Hepatic steatosis	High-fat diet-fed rat model	[83]
	Cardiac hypertrophy	Deoxycorticosterone/salt (DOCA/salt)-induced hypertension uninephrectomized rat model	[102]

**Table 2 molecules-24-04426-t002:** In vitro models for the anti-inflammatory effects of sesame lignans.

Compound	Mechanism of Action	Cell Line	References
Sesamin	Inhibition of LPS-induced TLR4 expression	BV-2 microglial cell	[34]
	Inhibition of LPS-induced IL-6 mRNA and protein	BV-2 microglial cell	[35]
	Inhibition of HO-1 protein ubiquitination	RAW 264.7 murine macrophage cells	[48]
	Activation of Nrf2/ARE	PC12 rat pheochromocytoma cells.	[49]
	Inhibition of IL-1β-stimulated human osteoarthritis chondrocytes.	Primary chondrocytes	[58]
	Inhibition of oxidized low-density lipoprotein (oxLDL)-induced endothelial dysfunction	Human umbilical vein endothelial cells (HUVECs)	[97]
Episesamin and Sesamin metabolites	Activation of Nrf2/ARE	PC12 rat pheochromocytoma cells.	[49]
Sesamin Catechol Glucuronides	Inhibition of LPS-induced NO production	J774.1 mouse macrophage-like cells	[75]
Sesamol	Inhibition of MMPs expression	SW1353 human chondrosarcoma cells	[59]
	Inhibition of inflammatory LOX	Soy LOX-1 enzyme model	[113]
Sesaminol Triglucoside	Inhibition of IL-6 and TNF-α	RAW 264.7 murine macrophage cells	[93]
Sesamolin	Reduce the activation of p38 MAPK	BV-2 microglial cell	[35]

#### 2.1.8. Others

The extent of the effect of other sesame extracts on the inflammatory cytokines and mediators has also been investigated thoroughly. One study involves the aqueous extract of sesame oil (SOAE) [114], which has twenty-eight identified molecules that ranges from moderate to polar in nature. The SOAE-methoxyphenol derivatives (SOAE-8; i.e., vanillyl alcohol, p-hydroxyphenylacetic acid, vanillic acid, coniferyl alcohol, p-coumaric acid, ferulic acid, sinapic acid and syringic acid) are the key components for its anti-inflammatory property. The absence of the sesame lignans in SOAE paves way for the study to be engrossing. Monocyte derived macrophages (MDMs) and RAW 264.7 macrophage cells were used, and the results are positive with slight distinction. SOAE-8 successfully reduced dose-dependently the mRNA levels of the inflammatory markers (*IL-6, IL-1β* and *TNF-α*) in MDMs. Meanwhile, in RAW264.7 cells, only *TNF-α* mRNA level was not reduced dose-dependently by SOAE-8.

The ethanol extract in black sesame seeds (BSSEE) attenuates liver inflammatory response in fructose-induced NAFLD rat model [115]. Three major lignans, which are sesamin (16.33%), sesaminol (1.92%) and sesamolin (13.06%), were found in BSSEE. Three major lignans, which are sesamin (16.33%), sesaminol (1.92%) and sesamolin (13.06%), were found in BSSEE. Inflammatory cytokines are dose-dependently reduced by BSSEE when administered at 0.5–2 ppm. Infiltration of inflammatory cells is also hindered in the presence of BSSEE. Correspondingly, BSSEE with concentrations of 1 ppm and 2 ppm promotes the activation of Nrf2 and improves the levels of MAPKs and NF-кB. In another study of BSSEE, Freund’s complete adjuvant (FCA)-induced arthritis rat model was used [116]. The study examined the effect of BSSEE on the inflammatory cytokines (IL-6 and TNF-α). It has been reported that at dosage 800 ppm of BSSEE, the levels of inflammatory cytokines induced by rheumatoid arthritis are reduced in the span of 28 days. Comparatively, extracts of black sesame seeds via CO_2_ supercritical fluid extraction (SFE) exhibits neuroprotective activity against ischemia [117]. While its composition is mainly made up of fatty acids (caprylic acid, capric acid, lauric acid, myristic acid, palmitoleic acid, margaric acid, linolenic acid, arachidic acid, behenic acid, palmitic acid, stearic acid, linoleic acid and oleic acid) and phytosterol (cholesterol, brassicasterol, stigmasterol, ∆-5, 24 stigmastadienol, ∆-7 stigmastanol, ∆-7 avenasterol, eritrodiol, campesterol + campestanol + 24 methylene cholesterol, clerosterol + ∆-5, 23 stigmastadienol, ∆-5 avenasterol and β-Sitosterol + sitostanol), its synergistic interactions play a vital role. The infiltration of leukocyte reduces when treated with CO_2_ SFE extracts of black sesame seeds. On the contrary, the study suggests further evaluation of its mediating-effect on the neuronal disorders.

The ethanol extract in the sesame coat (EESC) also manifests anti-inflammatory property [118]. It was reported that EESC contains sesamin, sesamolin, phenolic compounds and tetranortriterpenoids [119]. Using RAW 264.7 macrophage cell line, EESC (0.08 ppm) exhibited lowering effect on the levels of NO production as well as PGE_2_ production. EESC, at the same dosage, inhibits both the *iNOS* and *COX-2* protein expressions by 94% and 53%. Inhibitory effect on the LPS-induced degradation of IкB-protein has been observed when EESC was applied.

**Table 3 molecules-24-04426-t003:** In vitro/in vivo applications of different extracts from different sesame components.

Sesame Component	Mode of Extraction/Solvent	In vitro/In vivo	References
Sesame Oil	Aqueous extract	RAW 264.7 macrophage cell line	[114]
Black Sesame Seeds	Ethanol extract	Fructose-induced NAFLD rat model	[115]
Black Sesame Seeds	Ethanol extract	Freund’s complete adjuvant (FCA)-induced arthritis rat model	[116]
Black Sesame Seeds	CO_2_ supercritical fluid extraction	Endothelin-1-induced focal ischemia rat model	[117]
Sesame Coat	Ethanol extract	RAW 264.7 macrophage cell line	[118]

### 2.2. Anti-Cancer Activity

During inflammation, the immune system of the body is tasked to the release of reactive oxygen and nitrogen species (RONS) to fight against pathogens and to protect the body. RONS are also responsible for tissue repair and regeneration [120]. However, these chemicals are able to obstruct DNA repair mechanisms, which can potentially lead to DNA damage. With damaged DNA, the chance of mutations significantly rises, promoting tumorigenesis. It was reported that chronic inflammation is part of the 15% precedencies of recorded cancer cases [121]. Millions of cancer cases are recorded every year and millions of people succumb to different types of cancer. In 2018, 9.6 million deaths in the world were caused by cancer [122]. Due to this, efforts in finding a cure for and attempts to understand cancer continue to rise [123]. Cancer is commonly defined as the abnormal growth of the cells, but there are indications that mark the development of cancer in the human body. It was suggested that six essential alterations in cell physiology are mainly responsible for the uncontrolled growth of the cells. Self-sufficient in growth signals, insensitive to growth-inhibitory signals, evasive of apoptosis, unlimited replication potential, maintenance of angiogenesis and invasive to tissues are the six characteristics of cells during tumor development [124,125]. However, in a study more than a decade later after these indications were identified, two other hallmarks of cancer have been discovered. It was reported that tumor cells are also able to reprogram energy metabolism and are evasive of immune destruction, with genome instability and inflammation as the primary causes [126]. Knowing these hallmarks allow the creation of defensive strategies against cancer and also understanding the fundamentals of these indications opens the opportunity for a better comprehension as to how neoplastic diseases emerge.

#### 2.2.1. Lung Cancer

Oil extracted from sesame stood out among other vegetable oils because of its exceptional nutritional characteristics [2]. Sesame lignans have demonstrated several pharmacological applications [127], including anti-proliferative activity, which made them subjects of anticancer studies. Lung cancer, which is the most diagnosed cancer in 2018 [122], has been studied by Harikumar and colleagues by treating human lung adenocarcinoma cell line H1299 with sesamin solution. Sesamin was able to significantly inhibit the proliferation of H1299 cells, with a 50% inhibitory dose of 40.1 µmol/L [128]. Sesaminol has also been proven to exhibit anticancer property against lung carcinoma A549 cells at a concentration of 50 µM and 6 h of treatment [129]. The mechanism of action of sesaminol on A549 cells is tackled in the following section (2.2.2 Breast Cancer). 

Another lignan of sesame, sesamol, has also been explored for its anticancer properties [130]. It has been studied for its apoptotic effect in lung adenocarcinoma SK-LU-1 cell line. After 48 h of treatment, the lignan showed selective antiproliferation effect with an IC_50_ of 2.7 mM on SK-LU-1 cells and 7.6 mM on Vero cells. The apoptotic effect of sesamol was also found to be both time and dose-dependent. The higher the concentration, the higher number of cell deaths was recorded. At 2.7 mM sesamol, evident necrosis occurred in a time-dependent manner, while at 5.4 mM sesamol, early stage apoptosis was affected [131,132]. A 24-h sesamol treatment of SK-LU-1 cells showed an increase in the activities of caspase 3/7, which largely participate in the propagation of death signals. Specifically, the increase and activation of caspase 3 causes the cleavage of a DNA repair protein known as PARP. Consequently, DNA damage occurs and cell deaths are achieved. The loss of mitochondrial membrane potential (Δψm) was also investigated as one of the factors of apoptosis and a 48-h treatment revealed that longer exposure to sesamol leads to a greater decrease in Δψm. This proposes that mitochondria are part of the apoptotic pathway that occurs in SK-LU-1 cells [131,132].

Another study revealed that sesamin also has a protective action against the down-regulation of the PI3K-Akt signaling pathway in a nickel-induced apoptosis in mouse liver. PI3K-Akt pathway is responsible for the restriction of apoptosis and for the promotion of cell survival [133,134]. This action of sesamin showed that the lignan is capable of preventing DNA damage, hence, it becomes a potential anticancer agent. In line with this, it was also reported that the ability of sesamin to reduce *COX-2* gene expressions in A549 cell line occurs through PI3K-Akt pathway [135] and inhibition of this pathway inactivates inflammatory response, which reduces restenosis [136].

#### 2.2.2. Breast Cancer

Another common type of cancer is the female breast cancer, which accounts for 11.6% of the total cancer deaths [122]. For that reason, breast cancer is also widely studied. In 2007, sesamin was examined for its effect on the proliferation of human breast cancer cell line MCF-7 for 24 h. Results showed that the inhibition is done through G1 phase growth arrest and is dose-dependent, with a cytostatic effect at 100 µM sesamin. The lignan was also used to assess the effect of down-regulation of cyclin D1 protein expression in different types of human tumor cells including the breast cancer cell line T-47D, the lung cancer cell line A549, the transformed renal cell line 293T, the immortalized keratinocyte cell line HaCaT, the melanoma cell line UACC-62 and the osteosarcoma cell line MG63, which proved that suppression of sesamin generally occurs in the tumor cells. In line with this, the down-regulation of cyclin D1 protein expression was examined as one of the main factors that cause the growth inhibitory effect of sesamin against cyclin D1-depleted MCF-7 breast cancer cells. Results showed that the inhibitory effect of sesamin is largely dependent on the presence of cyclin D1 as cyclin D1-depleted MCF-7 cells were almost insensitive to the sesamin treatment [137]. In the study of Harikumar, the same antiproliferation activity of sesamin was also observed against breast cancer cells MDA-MB-231 with an IC_50_ value of 51.1 µmol/L [128].

The anticancer function of sesamin also manifests even with a different mechanism of action. Sesamin was also able to impede the proangiogenic activity of MCF-7 cells. A 24-h pretreatment of MCF-7 cells and macrophages were carried out with 50 µM of sesamin, which resulted to the suppression of angiogenesis upon observation of the endothelial capillary tube assay and the network formation of the cells. Observation done on MCF-7 cells alone, however, revealed that the sesamin treatment was not significantly cytotoxic to the breast cancer cells as it did not decrease the cells’ viability even at sesamin concentration of 100 µM within a 72-h treatment period. MФCM-treated MCF-7 cells were also examined for the same activity and sesamin showed a more promising inhibition performance than when MCF-7 cells were treated with the control medium. The same suppressive effect of sesamin was also exhibited against a more malignant cell line, MDA-MB-231 cells, treated with MФCM. To properly investigate the mechanism of action of sesamin on the tumor cells, vascular endothelial growth factor (VEGF) and matrix metallopeptidase 9 (MMP-9), which are essential factors of angiogenesis, were evaluated. MФCM treatment was found to induce *VEGF* and *MMP-9* mRNA expression in MCF-7 cells, but treatment with sesamin drastically hindered this. The same results were obtained for MDA-MB-231 cells [138].

A combined treatment with γ-tocotrienol also prevents the proliferation of MCF-7 and MDA-MB-231 cells. It was suspected that sesamin could increase the antiproliferative activity of γ-tocotrienol by inhibiting its metabolic degradation but the study revealed that the synergistic inhibiting effect of the two compounds is the result of the induction of G1 cell cycle arrest and the reduction of protein expression levels involved in the cell cycle. The synergistic inhibitory performance of sesamin and γ-tocotrienol is also effective against the murine +SA mammary epithelial cell line of a mouse [139]. An in vivo study on the proliferation of MCF-7 cells in athymic mice presented a comparison of the apoptotic activity of sesamin and of a lignan of flaxseed, secoisolariciresinol diglucoside (SDG), in which sesamin showed a more promising activity [140]. Tumor cell proliferation declined by 38% with sesamin treatment and cell apoptosis rose by 91% as opposed to the 37% reduction of proliferation with SDG treatment without exhibition apoptotic property. Their ability to down-regulate growth factor receptors known as EGFR and HER2 explains the reduced tumor growth and sesamin’s ability to reduce pMAPK expression causes it to be more effective than SDG.

Just like with sesamin, sesamol was also used against the human breast cancer MCF-7 cells. Endothelial cell line EA.hy 926 was used to observe morphological changes that are caused by the exposure to sesamol. Endothelial cells were treated with sesamol for 72 h with a concentration range of 0.05–1.0 mM. It was found that lower doses of sesamol caused the cells to swell, while a concentration of 1 mM caused apparent cell death. Fragmented nuclei were present in the cells that were treated with 1 mM sesamol, indicating apoptosis. To check whether sesamol is capable of inhibiting the growth of MCF-7 cells, the cells were subjected to a 3-day treatment at 0.10 mM. This revealed that cell numbers are indeed lower than the controls. Combining PI labeling with a TUNEL assay led to the information that cell deaths occur in both S and G_2_/M phases [141]. Whether the inhibitory effect of sesamol on MCF-7 cells was solely caused by the apoptosis was disclosed in the study. No other pathway of inhibition was presented. 

Aside from this, sesaminol has also been utilized against breast cancer cell lines MDA-MB-231 and MCF-7, along with lung carcinoma cell line A549 stated in the previous section. As mentioned, restriction of cyclin D1 expression plays a major role in some of the inhibitory pathways induced by sesame lignans. It was reported that the same statement applies to sesaminol, which was verified when a 6-h treatment at 50 µM sesaminol reduced cyclin D1 expression in MCF-7 cells. Although the same conditions did not produce the same result in MDA-MB-231 cells, it was stated that 100 µM sesaminol with 24 h to 48 h of sesaminol treatment achieved the desired result. To investigate how sesaminol manifests the same activity, sesaminol-immobilized FG beads were incubated with MCF-7 cells to identify, which proteins would bind to the beads. Among the proteins was an inner mitochondrial membrane protein known as adenine nucleotide translocase 2 or ANT2. This protein is said to be overexpressed in different malignant tumors and is recognized as an oncoprotein. To confirm whether ANT2 is involved in the anticancer activity of sesaminol, a knockdown of ANT2 in MCF-7 cells was performed. This resulted to growth inhibition and accumulation of cells in the G1 phase [129].

#### 2.2.3. Prostate Cancer

Following breast cancer is the prostate cancer, which is responsible for 7.1% of the total cancer deaths. Similar with the findings on γ-tocotrienol, γ-tocopherol was also found to have antiproliferation effect on the human prostate cancer cell line PC-3 [142] and that intake of sesamin, sesamolin or sesame seed oil in general [143] has a significant impact on the plasma levels of tocopherols in humans [144]. The presence of γ-tocopherol becomes more evident with sesamin intake and at low levels of α-tocopherol [145]. It was reported that sesamin prevents the tocopherol from being metabolized by inhibiting CYP3A-dependent carboxychroman production to which tocopherol is metabolized to [146]. The same lignan was also found effective against the solid tumor cell line DU145 and the 50% inhibitory sesamin dose was 60.2 µmol/L [128].

#### 2.2.4. Colon Cancer

Colorectal cancer, which is next to prostate cancer as one of the leading causes of cancer deaths [122], was investigated to determine whether sesame lignans can act against its cell lines. Sesamin was capable of suppressing the growth of colon cancer cells HCT116 with an IC_50_ value of 57.2 µmol/L [128] and in the study of Watanabe, colon adenocarcinoma RKO cells were also used to observe the effectivity of sesaminol against the proliferation of these cancer cells and the same results were observed with 6-h treatment of 50 µM sesaminol [129]. Five compounds found in sesame seeds, including sesamol, were tested for their ability to suppress the transcriptional activity of COX-2 because its excessive production of prostaglandin is an essential factor of colorectal cancer. The human colon cancer cell line DLD-1 was used to measure the activity of COX-2 up to 100 µM of the test compounds. At 100 µM, the inhibiting effect of sesamol was 50%, while ferulic acid, sesamin, sesamolin and syringic acid did not manifest any favorable results. Due to this, sesamol was further examined in terms of its capability to hinder intestinal polyp formation in Min mice. The effects of administering 500 ppm sesamol to Min mice for eight weeks were observed. It was stated that sesamol administration did not affect the mice in any aspect abnormally and that it successfully lessened the number of polyps in the small intestine and colon by 75% of the untreated group [147]. 

Human colorectal carcinoma cell line HCT116 was also evaluated as the target of sesamol, to which it caused cell death with an IC_50_ value of 2.59 mM. Concentration range 0.5–5 mM showed exceptional inhibition of the survival of HCT116, but not on the viability of the normal Vero cells. The cell cycle arrest ability of sesamol was then compared to cisplatin with an exposure of 48 h. Sesamol was able to induce G_0_/G_1_ cell cycle arrest at a low concentration (0.05 mM) as opposed to 100 µM of cisplatin. At both low and high concentrations, sesamol was also able to arrest the S phase, with the highest cell cycle arrest peaking at 1 mM [148]. Treatment beyond this concentration revealed a relationship with the S phase arrest that is inversely proportional. Sesamol increases the cells in the S phase and decreases the number in the G_0_/G_1_ phase. Analyzing this led to the understanding that subG1 phase is also suppressed by both sesamol and cisplatin, leading to DNA fragmentation and cell death. Data gathered translate to the fact that the mitochondria are indeed a factor in the apoptotic pathway induced by sesamol as it enhanced Δψm. Colon cancer induced by 1,2-dimethylhydrazine (DMH) in Wistar rats was observed to whole sesame paste (WSP) and resistant starch type 2 (RS2) as anticancer agents. It was reported that both WSP and RS2 have restrictive actions against the initiation of DMH-induced colorectal cancer and they are capable of reducing the number of mucin depleted foci [149]. This result is similar to the study on azoxymethane-induced colon carcinogenesis, which proved that sesame could act against the said cancer [150].

#### 2.2.5. Liver Cancer

The induction of cell cycle arrest was also tested on the human hepatocellular carcinoma cell line HepG2. The MTT assay was used to observe the viability of HepG2 cells under the influence of sesamin and the data indicated that the cells were inhibited after 48 h with an IC_50_ value of 98 µM, but sesamin was less cytotoxic to L02 cells. Unlike with the growth inhibition of the breast cancer cells, the antiproliferation activity on HepG2 cell was caused by the suppression of the STAT3 signaling pathway, which controls genes that participate both in cell cycle and apoptosis. This induces G2/M phase arrest and a dose-dependent early apoptosis resulting to reduced proliferation [151]. Parallel observations were made when sesamol was used against HepG2 cells, in which the antiproliferative activity of 1 mM sesamol was over 90%. Just like with other inhibitory effects of sesame lignans, this was also reported as concentration-dependent. To understand which cell death patterns occur in HepG2 cells, the deaths were observed at different concentrations. Chromosomal DNA fragmentation was noticeable at as low as 50 µM sesamol. At this concentration, characteristics of apoptosis such as nuclear shrinkage and membrane blebbing were recorded. At higher doses like 200 and 1000 µM, necrosis was observed [152,153].

The location of sesamol in cells was identified to further analyze its apoptotic effect. It was shown that sesamol undergoes nuclear localization in HepG2 cells and this phenomenon is related to sesamol’s cytotoxicity because this means that sesamol can travel to and accumulate in the nuclei of cancer cells. The transportation of sesamol into the nucleus, however, was not discussed in the study. In a more recent study of the same group of researchers, it was revealed that the intrinsic apoptosis pathway was achieved by the reduction of the Bcl-2 expression, although there were no changes observed on the expression of Bax. Sesamol also enhanced cytochrome *c* release, which activates caspase-3. As discussed previously, caspase-3 activation leads to the cleavage of PARP, which is responsible for DNA repair. In contrast, the extrinsic pathway happened through the amplified protein expression of Fas/FasL and through the activation of tBid and caspase-8, caused by sesamol. Further experimentations also showed that sesamol suppressed both autophagy and mitophagy in the liver cancer cells via reduced LC3 expression, an indicator of autophagy, and via triggering the loss of Δψm [152,153]. Phenolic extract from black and white sesame seeds have also been used to study the same activity and have been proven to exhibit significant antiproliferative property [154].

#### 2.2.6. Cervical Cancer

Cervical cancer, ranked fourth as the leading cause of cancer death among females [125], was studied by observing the effect of sesamin on the viability and the migration of the HeLa human cervical cancer cell line. The CCK-8 assay was used to determine the cells’ viability, while scratch wound assays were used for the migration test. Similar with other inhibitory actions of sesamin, its effect on the HeLa cells was also found to be dose-dependent. The apoptosis rate of HeLa cells also increased with 50 µM sesamin treatment within 48 h as compared to the HeLa cells without sesamin treatment and this occurrence was caused by the increased ratio of Bax, a pro-apoptotic protein, to Bcl-2, an anti-apoptotic protein [155]. It was also revealed that sesamin treatment increased the injured endoplasmic reticulum leading to programmed cell deaths. There was also up-regulation of the levels of p-IRE1α and p-JNK in HeLa cells, which were reported to be the pathway responsible for the ER-stress mediated apoptosis.

#### 2.2.7. Blood Cancer

Sesamin can also suppress the proliferation of human leukemic cell lines, KBM-5 and K562, and of a multiple myeloma cell line, U266. The IC_50_ values of sesamin for these cells are 42.7, 48.3 and 51.7 µmol/L, respectively. While investigating the biological pathway of sesamin in these cells, it was discovered that pretreatment with the lignan allows the restriction of cyclooxygenase-2 (COX-2) and cyclin D1 expressions induced by tumor necrosis factor (TNF), which is a cell signaling protein. These expressions are known to play essential roles in the propagation of cancer. Aside from this, TNF can also induce expression of gene products involved in angiogenesis and sesamin exhibited the same inhibitory action against these expressions. It was revealed that sesamin is capable of hindering the growth of the cells and its inhibition ability depends on both its concentration and the duration of the treatment. The inhibitory action of sesamin was made possible with the induction of TNF to NF-κB activation, which is responsible for the involved cellular responses. Sesamin was able to augment the apoptotic activity of TNF by downregulating the expression of gene products [131].

Aside from the study of Harikumar on leukemic cells using sesamin treatment, study utilizing sesamol against blood cancer cells have also been carried out. Interaction of sesamol with human lymphoid leukemia Molt 4B cells resulted to growth inhibition and induced apoptosis in a concentration-dependent manner. Morphological change indicating apoptosis and DNA fragmentation were observed in the cancer cells and the fragments of DNA increased the longer the contact time. These changes were not observed in sesamol-treated normal lymphocytes, leading to the conclusion that sesamol could induce cell death to restrict the growth of the leukemic cells [156]. The study did not present a detailed mechanism of action of the DNA fragmentation nor the apoptotic pathway. 

A study in 2010, both in vivo and in vitro, presented the cytotoxic activity of two oxidation products of sesamol, a trimer and a tetramer. Sesamol and its oxidation products were used to treat rat thymocytes to know how the compounds will change the lethality of the cells. FeCl_3_ was reacted with sesamol to undergo oxidation and yield the trimer and the tetramer. A 24-h incubation with 30 µM sesamol did not affect the population of the cells exerting propidium fluorescence, which is used as an indicator for dead cells. On the other hand, trimer at the same concentration resulted to a slight growth in population, while tetramer amplified it. The biological pathway of the apoptotic effect of tetramer was similar to that of sesamol against the SK-LU-1 cell line. Tetramer managed to elevate the activity of caspases, which in turn increased DNA damage. When tetramer was tested on K562 cells as an antiproliferative agent, the results showed that K562 cells were inhibited depending on the concentration of tetramer, which ranged from 3 to 30 µM. Only the concentration above 10 µM was able to manifest significant inhibitory performance. It was also discovered that at 30 µM, tetramer already exhibits a significant increase in lethality. Unfortunately, at these specific concentrations, the cytotoxicity of tetramer on normal cells was found to be greater than the cancer cells, which makes it difficult to consider the compound as a possible anticancer agent [157]. To be able to utilize tetramer against K562 cells, a mechanism of action that protects the normal cells from its cytotoxic action must first be developed. Acute myeloid leukemia cells, HL-60 and Molt-4, were also examined and only HL-60 cells suffered from DNA fragmentation due to exposure to sesamol [158].

The idea of sesaminol acting against human lymphoid leukemia Molt 4B cell line was also investigated by the same research team that explored sesamol’s effect on Molt 4B cells and parallel data were acquired as with the study on sesamol. The inhibition was also concentration-dependent and morphological changes indicating apoptosis were also reported. Apoptotic bodies were observed after three days of treatment with 45 µM sesaminol and it was noted that the growth inhibition of sesaminol is better than other sesame lignans. Specific mechanism of action of sesaminol on its induced apoptosis in Molt 4B cells was also not presented in this study [159,160].

On the other hand, along with sesamin, sesamolin was studied for its inhibitory effect against Burkitt’s lymphoma cells, Raji. The study aimed to utilize the sesame lignans to improve NK cell lysis activity so identification of a cancer cell line that has low cytotoxicity against NK cells was first carried out. Human leukemia cell line K562, T cell leukemia cell line Jurkat, and human Burkitt’s lymphoma cell line Raji were the three tumor cell lines used with LHD assay. Both K562 and Jurkat cells were highly sensitive to the cytolysis activity of the NK cells in contrast with its effect on Raji cells. Due to this, Raji cell line was used as the subject of study. It was shown that sesamolin decreased the cells’ viability by 31% compared to the untreated cells at a concentration of 80 µg/mL. Meanwhile, sesamin showed an even greater cytotoxic activity with concentrations higher than 2.5 µg/mL. Since the goal of the study was to use the lignans as boosting agents, concentrations were set at a level that is not toxic to Raji cells. Sesamolin-treated Raji cells were proven to be more sensitive to NK cell lysis than the untreated ones, confirming that sesamol enhanced the lysis activity of the NK cells. On the other hand, no significant results were observed with the tests involving sesamin. Since NKG2D ligands are deemed as one of the key role players in the activity of NK cells against cancer cells, it was assessed whether sesamolin treatment has an impact on its expression. Below toxic concentration of sesamolin, it was confirmed that the expression indeed increased. The same amplification of expression was observed with other key role players namely ULBP-1, ULPB-2 and MIC-A/B. The latter part of the study reported that sesamolin increased the phosphorylation of the ERK pathway, which is one of the pathways responsible for NKG2D ligand expression in Raji cells [161]. Although sesamin did not affect the NK cell lysis activity, it is important to note that it exhibits a much greater cytotoxicity to Raji cells even at low concentrations. This presents the possibility of sesame acting against the growth of Raji cells in a different pathway. 

#### 2.2.8. Skin Cancer

The antimelanogenesis property of sesamin, was evaluated in comparison to recognized depigmenting agents, kojic acid and β-arbutin. Its ability to function as a sunscreen was investigated through the measurement of UV absorption. Its effects against tyrosinase activity through mushroom and cellular tyrosinase were also observed. The results showed that sesamin was able to absorb ultraviolet in the UVB range with an absorbance intensity of approximately 1.3 at 290 nm, as opposed to the absorbances of kojic acid and β-arbutin equal to 0.2 at 300 nm. Sesamin also exhibited a slight inhibitory effect against tyrosinase. In the same study above, similar antimelanogenesis findings were observed for sesamol. The lignan was also able to absorb ultraviolet in the UVB range with an absorbance intensity of 0.8 at 300 nm, four folds higher than that of kojic acid and β-arbutin. Its inhibition of mushroom tyrosinase activity in vitro was shown to be concentration-dependent just like with kojic acid. The IC_50_ values for sesamol and kojic acids were 0.33 µg/mL and 6.15 µg/mL, respectively. On the other hand, β-arbutin did not exhibit an inhibitory activity, leading to the conclusion that sesamol has the greatest effect against tyrosinase enzyme [162].

To further assess the ability of sesamol, its cytotoxicity on Vero and SK-MEL2 cell lines were examined. Cytotoxicity of sesamol to Vero cell line was 22.8% after 48 h of treatment, while at a concentration range of 600–800 µg/mL, its cytotoxicity to SK-MEL2 cell line was between 42–97%, with an IC_50_ value of 608.93 µg/mL. Sesamolin, another lignan, was also tested for its antimelanogenesis activity. Similar with the results in the inhibition of mushroom tyrosinase activity, the sesame lignans and kojic acid obstructed the activity of cellular tyrosinase, while β-arbutin was negative. In line with this, sesamolin was also found to manifest the highest inhibition equal to 50% compared to the other two sesame compounds with only 23% of inhibition. Inhibition of melanin pigment was also investigated among the test compounds and sesamolin showed the most favorable inhibition performance at 25 µg/mL. Other proteins involved in melanogenesis, TRP-1 and TRP-2, were also exposed to the sesame compounds and their levels in SK-MEL2 cells were analyzed using Western blot analysis. It was illustrated that sesamolin lowered TRP-1 and TRP-2 protein levels by 36% and 15%, respectively. Sesame oil was also tested against SK-MEL cells [162], while sesaminol was tested against the SK-MEL-28 cell line, which was found to reduce cyclin D1 expression at a concentration of 100 µM within one to two days of incubation period [132].

Studies have shown that essential polyunsaturated fatty acids such as linoleic acid are capable of exhibiting antiproliferative activity against malignant cell lines. It was also reported that sesame oil is made up of 96% triglycerides and around 90% of its esterified fatty acids are oleic and linoleic acids in an approximately equal proportion. In line with this, sesame oil was proven to manifest greater growth inhibitory effect against melanoma cells than compared to its effect on normal melanocytes. An incubation period of five days showed that the average growth rate of melanoma cells and normal cells were 2.6 and 2.2, respectively [163]. Further investigation on the pathway of inhibition, however, was not executed. In a similar study, the effect of sesamin, sesamol and sesamolin on the synthesis of melanin in mouse melanoma B16F10 cells was observed and only sesamol showed a significant inhibitory effect, which was approximately 63% of the synthesis at 100 µg/mL. The production of melanin was halted by sesamol by inhibiting the specific activities of mushroom tyrosinase, monophenolase and diphenolase. At the same concentration, sesamol was also able to disrupt the viability of the cells by 60% as it was found to exhibit an apoptotic effect [164].

The chemopreventive capability of sesamol, along with sesame oil and two other products, against two-stage skin carcinogenesis of mice was also inspected in vivo. The two-stage skin carcinogenesis initiated by 7,12-dimethylbenz(a)anthracene (DMBA) and promoted by the tumor promoter 12-*O*-tetradecanoylphorbol-13-acetate (TPA) was exposed to resveratrol, sesamol, sesame oil and sunflower oil. Ten weeks of treatment and prior to TPA promotion, the mice showed gross tumor incidence of 20%, 20%, 30% and 20%, respectively, against the 100% control group. After 20 weeks of treatment, only resveratrol and sesamol manifested more than 30% and 10% inhibitory potential, respectively. The ability of the compounds to cancel TPA tumor promotion and impede tumor latency further demonstrated their chemopreventive effects [165]. In-depth mechanisms of action of the products involved against the proliferation of skin papillomas in the mice have not been provided. To test sesamol’s impact on the process of neoplastic development, an ex vivo research on the permeation of sesamol to the LACA mice skin was assessed by using sesamol solution and cream base sesamol-loaded solid lipid nanoparticles (S-SLN), while an in vivo method utilized sesamol solution, sesamol ointment and S-SLN. Ex vivo skin permeation of sesamol as a free drug solution was recorded to be much higher than that of S-SLN, which has three times higher skin retention and a 40% drop on the flux. This proposes that SLN is a potential transport service for sesamol. This is consistent with the results of the in vivo study wherein the number of papillomas on the dorsal skin of the mice was checked and there was 0% incidence of skin tumors [158].

#### 2.2.9. Others

Attempts to apply sesamin against pancreatic cancer and skin cancer were also executed. In 1994, pancreatic carcinogenesis initiated with *N-*nitrosobis (2-oxoproyl) amine (BOP) was observed in vivo by controlled diet of Syrian golden hamsters. The effect of sesamin-supplemented diet, a strategy to lower cholesterol levels, on the progress of the pancreatic cancer was evaluated. However, within a four-month period of treatment, it was concluded that although sesamin successfully lowered the cholesterol contents, this did not have any significant effect on BOP-initiated pancreatic cancer in hamsters [166,167]. Even so, the study emphasizes that the period of observation was done on a short term, which basis and suggests that favorable results possibly occur in a time-dependent manner. In line with this, a more recent study reported that sesamin was able to suppress the growth of pancreatic cancer MiaPaCa-2 cells with an IC_50_ value of 58.3 µmol/L [129].

The antitumor effect of sesamol was also determined using MA-10 cells, a mouse Leydig tumor cell line. Morphological changes caused by sesamol were observed and cells without sesamol treatment showed signs of normal cell growth phenomena, while those treated with sesamol appeared differently, depending on the duration of the treatment. Plasma membrane blebbing was seen after 12 h of treatment. These findings suggest the apoptotic property of sesamol. To confirm this, MA-10 cells viability against sesamol treatment was assessed. It was then revealed that sesamol is capable of inducing the death of MA-10 cells and that the dose of sesamol and the time of treatment are significant factors that affect the performance of sesamol. Flow cytometry showed DNA fragmentation at the subG1 phase of the cell cycle and verified the apoptosis of MA-10 cells. Similar with the established pathways of other studies, the activation of caspase-3 expression was also observed, consequently inducing apoptosis [168].

The following table (Table 4) provides a summary of the anticancer activities of the lignans of sesame, including their mechanisms of action against each cancer cell line discussed above:

## 3. Conclusions

The bioactive components of *Sesamum indicum* L., such as sesamin, sesaminol, sesamol and sesamolin, play essential roles in combating different types of biological and pharmacological concerns and are able to exhibit promising medicinal properties against the diseases. One of the notable properties of sesame lignans is anti-inflammation. Inflammation is a defense mechanism of a body against foreign substances and chronic inflammation persist to different kind of diseases. Sesame lignans hinder the propagation of inflammatory cytokines and inflammatory mediators, which further leads to alleviating inflammatory-related diseases such as osteoarthritis, cardiovascular disease, neurodegenerative disease, inflammatory bowel disease, diabetic eye disease, lung disease and liver disease. Other sesame extracts apart from the sesame lignans also exhibit mitigation of inflammatory-related pathways. These are proven by the in vivo and in vitro models of inflammatory-induced systems. Another alarming complication that can be instigated by inflammation is cancer. Cancer, which is the abnormal growth of the cells, may develop when the body suffers from DNA damage caused by the chemicals released during inflammation. Interestingly, it was also revealed that the lignans of sesame manifest anticancer activities against different cancer cell lines with different mechanisms of action. Both in vitro and in vivo studies have presented that the lignans are capable of inhibiting the growth of the cancer cells by down-regulating protein expressions, by suppressing the production of gene products, and through cell cycle arrest. Consequently, the lignans also induce either necrosis or apoptosis to the cells, inflicting an antiproliferation effect. Sesame lignans have been proven to manifest these anticancer effects against the tumor cells of lung cancer, breast cancer, colon cancer, prostate cancer, cervical cancer, blood cancer, skin cancer and even pancreatic cancer. With all things considered, sesame proves that food can indeed become a medicine and that foods do not only possess nutritional value, but they also have medicinal worth. The lignans of sesame that manifest anti-inflammatory and anticancer properties and the specific diseases that they act against are summarized in Figure 2 below:

## Figures and Tables

**Figure 1 molecules-24-04426-f001:**
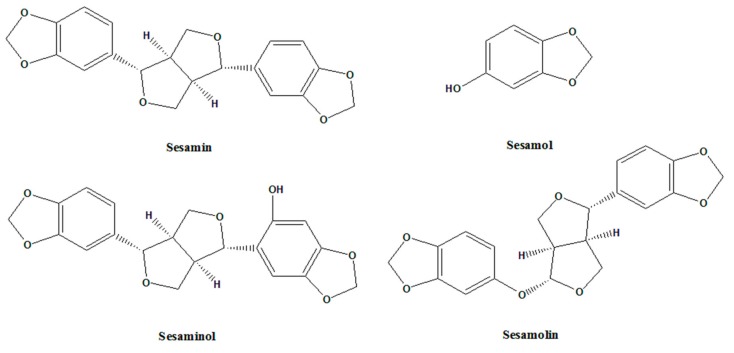
Chemical Structures of *Sesamum indicum* L. lignans.

**Figure 2 molecules-24-04426-f002:**
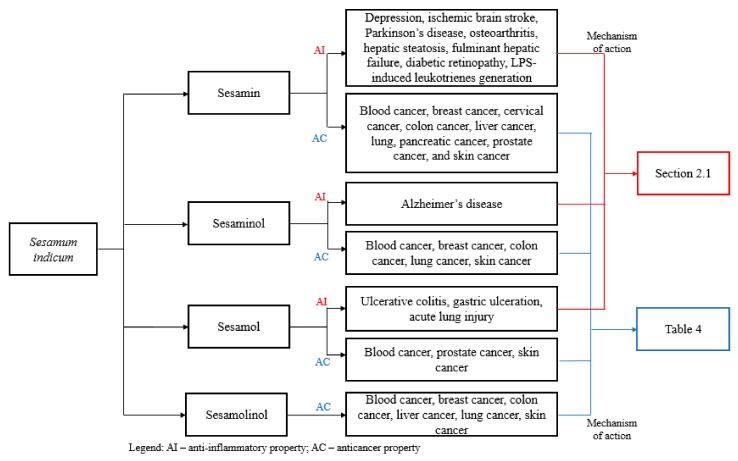
Anti-inflammatory and anti-cancer activities of major sesame lignans.

**Table 4 molecules-24-04426-t004:** Sesame lignans against different types of cancer and the mechanisms of action.

Compound	Cancer Type	Cell Line	Mechanism of Action	Reference
Sesame Oil	Skin Cancer			
	human malignant melanoma	SK-MEL	-	[163]
	two-stage mouse skin carcinogenesis	in vivo	protection against TPA tumor promotion	[165]
Sesame Extract	Liver Cancer			
	human hepatocellular carcinoma	HepG2	-	[154]
Sesamin	Blood Cancer			
	chronic myeloid leukemia	K562 KBM-5	inhibition of TNF-induced NF-κB activation	[128]
	myeloma	U266	inhibition of TNF-induced NF-κB activation	[128]
	Breast Cancer	MCF-7	induction of G1 cell cycle arrest; down-regulation of Cyclin D1 protein	[137]
		MCF-7 MDA-MB-231	inhibition of macrophage-induced VEGF and MMP-9 mRNA expressions	[138]
		MDA-MB-231	inhibition of TNF-induced NF-κB activation	[128]
		MCF-7 MDA-MB-231	induction of G1 cell cycle arrest and reduction of protein expression levels	[139]
		MCF-7	down regulation of growth factor receptors EGFR, HER2, and pMAPK expression	[140]
	Cervical Cancer	HeLa	favored apoptosis through the increase of Bax/Bcl-2 ratio; ER stress-mediated apoptosis by IRE1α/JNK pathway	[155]
	Colon Cancer			
	colon carcinoma	HCT116	inhibition of TNF-induced NF-κB activation	[128]
	Liver Cancer			
	human hepatocellular carcinoma	HepG2	suppression of the STAT3 signaling pathway	[151]
	Lung Cancer			
	human lung adenocarcinoma	H1299	inhibition of TNF-induced NF-κB activation	[128]
	Pancreatic cancer	MiaPaCa-2	inhibition of TNF-induced NF-κB activation	[128]
	Prostate Cancer	PC-3	degradation of γ-tocopherol metabolism	[146]
		DU145	inhibition of TNF-induced NF-κB activation	[128]
	Skin Cancer			
	malignant melanoma	SK-MEL2	absorption of ultraviolet in the UV range and inhibition of mushroom and cellular tyrosinase	[162]
Sesaminol	Blood Cancer			
	lymphoid leukemia	Molt 4B	DNA fragmentation leading to apoptosis	[159,160]
	Breast Cancer	MCF-7 MDA-MB-231	Reduction of cyclin D1 expression by binding to ANT2 protein	[129]
	Colon Cancer			
	colon carcinoma	RKO	Reduction of cyclin D1 expression by binding to ANT2 protein	[129]
	Lung Cancer			
	lung adenocarcinoma	A549	Reduction of cyclin D1 expression by binding to ANT2 protein	[129]
	Skin Cancer			
	melanoma	SK-MEL-28	Reduction of cyclin D1 expression by binding to ANT2 protein	[129]
Sesamol	Blood Cancer			
	acute myeloid leukemia	HL-60	DNA fragmentation leading to apoptosis	[158]
	chronic myeloid leukemia	K562	oxidation to tetramer; increased caspases activity leading to DNA damage	[157]
	lymphoid leukemia	Molt 4B	DNA fragmentation leading to apoptosis	[156]
	Breast Cancer	MCF-7	growth inhibition and apoptosis in S and G_2_/M phases	[141]
	Colon Cancer			
	colon adenocarcinoma	DLD-1	suppression of cyclooxygenase-2 transcriptional activity	[147]
	colon carcinoma	HCT116	subG1 phase cell cycle arrest causing cell death	[148]
	Liver Cancer			
	human hepatocellular carcinoma	HepG2	induced apoptosis and necrosis via DNA fragmentation and	[152,153]
			induced apoptosis via suppression of autophagy	
	Lung Cancer			
	lung adenocarcinoma	SK-LU-1	increased activity of caspase 3 leading to DNA damage	[131,132]
	Skin Cancer			
	human malignant melanoma	SK-MEL2	absorption of ultraviolet in the UV range and inhibition of mushroom and cellular tyrosinase	[162]
	mouse melanoma	B16F10	inhibition of monophenolase and diphenolase activities and promotion of apoptosis	[164]
	two-stage mouse skin carcinogenesis	in vivo	protection against TPA tumor promotion	[165]
	Others			
	Mouse Leydig tumor	MA-10	increased activity of caspase 3 leading to DNA damage at subG1 phase	[168]
Sesamolin	Blood Cancer			
	Burkitt’s lymphoma	Raji	enhancement of NK cell lysis activity via escalated NKG2D ligand expression	[161]
	Prostate Cancer	PC-3	degradation of γ-tocopherol metabolism	[146]
	Skin Cancer			
	human malignant melanoma	SK-MEL2	absorption of ultraviolet in the UV range and inhibition of mushroom and cellular tyrosinase	[162]

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
