# Peer review of "Anti-Inflammatory and Anticancer Properties of Bioactive Compounds from Sesamum indicum L.—A Review"

_molecules, 2019, doi:10.3390/molecules24244426_

Round 1
Reviewer 1 Report
The current work provides a comprehensive review of Anti-inflammatory and Anticancer Properties of Bioactive Compounds from Sesamum indicum L. Overall, it is nicely written. However, there are a few concerns:
The authors have mentioned 6 hallmarks of cancer from 2000 paper (ref 123). However, the latest review adds more hallmarks “Hallmarks of Cancer: The Next Generation” (2011). This needs to be updated. Provide a small cartoon for the representative properties of the compound.Author Response
REPLY:
Thank you for your suggestions. The update has been discussed from line 434 to line 437 and the study has been added as a new reference [126]. The diagram of both the anti-inflammatory and anti-cancer properties of the compound can be found from line 806 to line 809 (Figure 2).
Reviewer 2 Report
The review paper under consideration describes the anti-inflammatory and anticancer properties of S. indicum metabolites, mainly lignans.
The review is generally well organised and, despite some small language mistakes, is also well written and of clear comprehension. I understand Authors have focussed their attention to data related to pure compounds (sesame, sesame, etc..), however I would suggest to introduce also some more information about the use of standardised extracts. Thus, chemical composition of extracts reported in Table 3 should be detailed. The same applies to "sesame oil" which is mentioned in several paragraphs.
Author Response
REPLY:
Thank you for your suggestion regarding the chemical composition of each extract. We believe that the chemical composition of the aqueous extract of sesame oil was written already, hence we provided the chemical composition for BSSEE, extracts of black sesame seeds via CO2 SFE, EESC, and sesame oil in the new manuscript.
Line 61-63: Presence of sesame lignans such as sesamin, sesamol, sesamolin and other acylglycerols (oleic acid, linoleic acid, palmitic acid, stearic acid and arachidic acid) in sesame oil contributes to its distinct properties [19-20].
Line 392-393: Three major lignans which are sesamin (16.33%), sesaminol (1.92%) and sesamolin (13.06%) were found in BSSEE.
Line 403-408: While its composition is mainly made up of fatty acids (caprylic acid, capric acid, lauric acid, myristic acid, palmitoleic acid, margaric acid, linolenic acid, arachidic acid, behenic acid, palmitic acid, stearic acid, linoleic acid and oleic acid) and phytosterol (cholesterol, brassicasterol, stigmasterol, ∆-5, 24 stigmastadienol, ∆-7 stigmastenol, ∆-7 avenasterol, eritrodiol, campesterol + campestanol + 24 methylene cholesterol, clerosterol + ∆-5, 23 stigmastadienol, ∆-5 avenasterol, and β-Sitosterol + sitostanol), its synergistic interactions play a vital role.
Line 411-413: It was reported that EESC contains sesamin, sesamolin, phenolic compounds and tetranortriterpenoids [119].
Reviewer 3 Report
In this MS Ming-Shun Wu et al., review the current knowledge of sesamin anti-cancer and anti-inflammatory effects. The review is very informative and well written. Minor points I found that deserve attention by the Authors:
Several findings reported that sesamin inhibited PI3K/Akt signalling. The Reviewer suggests a careful revision of the issue regarding the cell signalling pathway like PI3K/Akt involved in anti-cancer and anti-inflammatory effects. This should be at least included in the introduction and/or discussion.
doi: 10.1042/BSR20171112.
doi: 10.1007/978-1-4939-1346-6_13
doi: 10.1016/j.ejphar.2019.05.008.
doi: 10.3892/ijmm.2018.3939.
doi: 10.1016/j.ejphar.2017.10.020.
Author Response
REPLY:
Thank you for your suggestions. The involvement of PI3K/Akt signalling pathway have been discussed in line 465 to line 471. The study with doi: 10.1007/978-1-4939-1346-6_13 was not included because the results of the study cannot be accessed by the authors. Another reference, which represents the effect of sesamin on the PI3K/Akt pathway, was added: doi: 10.1021/jf304562b ref [134]. The review paper (doi: 10.1016/j.ejphar.2019.05.008.) has already been part of the initial list of references (ref. 130).
Ref. [133]- doi: 10.1016/j.ejphar.2017.10.020.
Ref. [135]- doi: 10.3892/ijmm.2018.3939.
Ref. [136]- doi: 10.1042/BSR20171112.
Reviewer 4 Report
Authors reviewed anti-inflammatory and anticancer activities of lignans from Sesanun indicum L. Readers of Molecules are interested in this review. I recommend this review is to be published in Molecules.
The caption of Figure 1. Structures of Sesanun indicum L. lignans. I am afraid the absolute configuration of sesamolin is wrong. Please confirm the configuration of sesamolin. Page 9, Line 358. "the said proinflammatory mediator" Is this correct? Page 12, Line 399. "The ethanol extract in the sesane coat (EESC) also manifests.."Author Response
Thank you for your review. We also agree that the total configuration of sesamolin is incorrect, hence we re-draw the chemical structure of sesamolin to its correct configuration. (Line 68)
Line 362: the said inflammatory mediators.
Line 411: The ethanol extract in the sesame coat (EESC) also manifests
Round 2
Reviewer 1 Report
Comments have been addressed.